**PLOS** NEGLECTED TROPICAL DISEASES

# Screening for Chagas disease from the electrocardiogram using a deep neural network

**Carl Jidling**[1], **Daniel Gedon**[1], **Thomas B. Schön**[1], **Claudia Di Lorenzo Oliveira**[2], **Clareci Silva Cardoso**[2], **Ariela Mota Ferreira**[3], **Luana Giatti**[4], **Sandhi Maria Barreto**[4], **Ester C. Sabino**[5], **Antonio L. P. Ribeiro**[6], **Antônio H. Ribeiro**[1]*

**1** Department of Information Technology, Uppsala University, Uppsala, Sweden, **2** Preventive Medicine, School of Medicine, Universidade Federal de São João del-Rei, Divinópolis, Brazil, **3** Graduate Program in Health Sciences, Universidade Estadual de Montes Claros, Montes Claros, Brazil, **4** Preventive Medicine, School of Medicine, Clinical Hospital/EBSERH, Universidade Federal de Minas Gerais, Belo Horizonte, Brazil, **5** Instituto de Medicina Tropical da Faculdade de Medicina, Universidade de São Paulo, São Paulo, Brazil, **6** Department of Internal Medicine, Faculdade de Medicina, Telehealth Center, Hospital das Clínicas, Universidade Federal de Minas Gerais, Belo Horizonte, Brazil

* antonio.horta.ribeiro@it.uu.se

**Data Availability Statement:** SaMi-Trop cohort ECG tracings are available at https://doi.org/10.5281/zenodo.4905618. A fraction of the CODE data, the CODE-15% cohort, is openly available at

## Abstract

### Background

Worldwide, it is estimated that over 6 million people are infected with Chagas disease (ChD). It is a neglected disease that can lead to severe heart conditions in its chronic phase. While early treatment can avoid complications, the early-stage detection rate is low. We explore the use of deep neural networks to detect ChD from electrocardiograms (ECGs) to aid in the early detection of the disease.

### Methods

We employ a convolutional neural network model that uses 12-lead ECG data to compute the probability of a ChD diagnosis. Our model is developed using two datasets which jointly comprise over two million entries from Brazilian patients: The SaMi-Trop study focusing on ChD patients, enriched with data from the CODE study from the general population. The model's performance is evaluated on two external datasets: the REDS-II, a study focused on ChD with 631 patients, and the ELSA-Brasil study, with 13,739 civil servant patients.

### Findings

Evaluating our model, we obtain an AUC-ROC of 0.80 (CI 95% 0.79-0.82) for the validation set (samples from CODE and SaMi-Trop), and in external validation datasets: 0.68 (CI 95% 0.63-0.71) for REDS-II and 0.59 (CI 95% 0.56-0.63) for ELSA-Brasil. In the latter, we report a sensitivity of 0.52 (CI 95% 0.47-0.57) and 0.36 (CI 95% 0.30-0.42) and a specificity of 0.77 (CI 95% 0.72-0.81) and 0.76 (CI 95% 0.75-0.77), respectively. Additionally, when considering only patients with Chagas cardiomyopathy as positive, the model achieved an AUC-ROC of 0.82 (CI 95% 0.77-0.86) for REDS-II and 0.77 (CI 95% 0.68-0.85) for ELSA-Brasil.

https://doi.org/10.5281/zenodo.4916206. The data sets contain information about Chagas condition mortality, age, sex, the ECG tracings, and the flag indicating whether the ECG tracing is normal. The DNN model parameters that give the results presented in this paper were made available at https://doi.org/10.5281/zenodo.7371623. This should allow the reader to partially reproduce the results presented in the paper. Restrictions apply to additional clinical information on the CODE-15% and SaMi-Trop cohorts; to the full CODE cohort; to the REDS-II dataset and, to the ELSA-Brasil cohort. Researchers affiliated with educational or research institutions might make requests to access the data sets. Requests should be made to the Telehealth Center of Minas Gerais (address: Av. Professor Alfredo Balena, 110 - 1˚ Andar - Ala Sul - Sala 107 30130-100 - Belo Horizonte - MG, Brazil, email: telessaude@hc.ufmg.br). They will be forwarded and considered on an individual basis by the Telehealth Network of Minas Gerais, SaMi-Trop and ELSA-Brasil Steering Committee. An estimate for the time needed for data access requests to be evaluated is three months. If approved, any data use will be restricted to non-commercial research purposes. The data will only be made available on the execution of appropriate data use agreements. Datasets: [SaMi-Trop] Cardoso CS, Sabino EC, Oliveira CDL, de Oliveira LC, Ferreira AM, Cunha-Neto E, et al. Longitudinal study of patients with chronic Chagas cardiomyopathy in Brazil (SaMi-Trop project): a cohort profile. BMJ Open. 2016;6 (5). doi: 10.1136/bmjopen-2016-011181. [CODE] Ribeiro ALP, Paixao GMM, Gomes PR, Ribeiro MH, Ribeiro AH, Canazart JA, et al. Tele-electrocardiography and bigdata: The CODE (Clinical Outcomes in Digital Electrocardiography) study. Journal of Electrocardiology. 2019;57:S75–S78. doi: 10.1016/j.jelectrocard.2019.09.008. [REDS-II] Nunes MCP, Buss LF, Silva JLP, Martins LNA, Oliveira CDL, Cardoso CS, et al. Incidence and Predictors of Progression to Chagas Cardiomyopathy: Long-Term Follow-Up of Trypanosoma cruzi-Seropositive Individuals. Circulation. 2021;144(19):1553–1566. doi: 10.1161/CIRCULATIONAHA.121.055112. [ELSA-Brasil] Aquino EML, Barreto SM, Bensenor IM, Carvalho MS, Chor D, Duncan BB, et al. Brazilian longitudinal study of adult health (ELSA-Brasil): Objectives and design. American Journal of Epidemiology. 2012;175(4):315–324. doi: 10.1093/aje/kwr294. The code for the model training, evaluation and statistical analysis is available at the GitHub repository https://github.com/carji475/ecg-chagas.

**Funding:** CJ, DG, TBS and AHR are financially supported by the Wallenberg AI, Autonomous

## Interpretation

The neural network detects chronic Chagas cardiomyopathy (CCC) from ECG–with weaker performance for early-stage cases. Future work should focus on curating large higher-quality datasets. The CODE dataset, our largest development dataset includes self-reported and therefore less reliable labels, limiting performance for non-CCC patients. Our findings can improve ChD detection and treatment, particularly in high-prevalence areas.

## Author summary

Chagas disease (ChD) is a neglected tropical disease, and the diagnosis relies on blood testing of patients from endemic areas. However, there is no clear recommendation on how to select patients for testing in endemic regions. Since most cases of Chronic ChD are asymptomatic, the diagnostic rates are low, preventing patients from receiving adequate treatment.

The Electrocardiogram (ECG) is a widely available, low-cost exam, often available in primary care settings. We present an Artificial intelligence (AI) model for automatically detecting ChD from the ECG. AI algorithms have allowed the detection of hidden conditions on the ECG and, to the best of our knowledge, this is the first study that does it for ChD. We utilize large cohorts of patients from the relevant population of all-comers in affected regions in Brazil to develop a model for ChD detection that is then validated on datasets with ground truth labels obtained directly from the patients' serological status.

Our findings demonstrate a promising AI-ECG-based model for discriminating patients with chronic Chagas cardiomyopathy (CCC). The capacity of detecting ChD patients without CCC is still limited. But we believe this can be improved with the addition of epidemiological questions, and that such models can become useful tools for pre-selecting patients for further testing.

## Introduction

Worldwide it is estimated that Chagas disease (ChD) infects more than 6 million people, with thousands of deaths each year [1]. Caused by the protozoan parasite *Trypanosoma cruzi* (T. cruzi), the disease is endemic to countries in continental Latin America, but migration has carried ChD to new regions, including Europe and the United States [2]. The most critical consequence of ChD is chronic Chagas cardiomyopathy (CCC), which occurs in 20–40% of the infected individuals [3]. CCC comprises a wide range of manifestations, including heart failure, arrhythmias, heart blocks, sudden death, thromboembolism, and stroke [1, 3].

ChD is often a lifelong infection in which most chronically infected patients remain asymptomatic but at risk of progression to cardiac damage [4, 5]. The incidence of cardiomyopathy in those in this asymptomatic (indeterminate) form of ChD varies from 0.9 to 7% new cases annually [1] and is related to the parasite burden [5, 6]. There is no single gold-standard laboratory test for diagnosing chronic Chagas disease. Instead, at least two serological tests with different methods for detecting antibodies to *T. cruzi* and complementary sensitivity and specificity are needed to confirm infection [1, 3]. Treatment with antitrypanosomal drugs such as benznidazole can prevent progression to the cardiac form [7, 8], but it does not seem to prevent death and cardiac complications in those with advanced cardiomyopathy [9]. Thus, the early recognition of chronic ChD patients is a necessary step for treatment in the early phases,

Systems and Software Program (WASP) funded by Knut and Alice Wallenberg Foundation, and by the Kjell och Märta Beijer Foundation. ALPR is supported in part by CNPq (465518/2014-1, 310790/2021-2 and 409604/2022-4) and by FAPEMIG (PPM-00428-17, RED-00081-16 and PPE-00030-21). LG, SMB and the ELSA-Brasil study were supported by the Brazilian Ministries of Health and of Science and Technology (grants 01060010.00RS, 01060212.00BA, 01060300.00ES, 01060278.00MG, 01060115.00SP, and 01060071.00RJ). ECS, ALPR, CSC, CLO, AMF and the SaMi-Trop and REDS-II cohort studies were supported by the National Institutes of Health (P50 AI098461-02, U19AI098461-06, 1U01AI168383-01). LG, SMB, ECS and ALPR receive unrestricted research scholarships from CNPq; The funders had no role in the study design; collection, analysis, and interpretation of data; writing of the report; or decision to submit the paper for publication.

**Competing interests:** The authors have declared that no competing interests exist.

when treatment success rates are higher and can prevent severe organ damage from occur [10].

Even if the newly diagnosed patient has established cardiomyopathy, an early diagnosis will allow the initiation of guideline-directed medical therapy for clinical conditions, such as heart failure and atrial fibrillation, to halt disease progression and eventually prevent death [10]. ChD patients generally have low socio-economical levels and limited access to health services, and they frequently do not realize that they are infected. The awareness of ChD among healthcare providers is also low, and there is a lack of knowledge on who to screen as well as a lack of clarity on the appropriate tests and clinical management [11, 12].

In many countries, there are detection rates below 10%, even more frequently, below 1%. The low detection rates create a barrier to the health care system, preventing patients from receiving adequate treatment [13]. The under-appreciation of early diagnosis and treatment, especially at the primary healthcare level, represents a missed opportunity for modifying the natural history of the disease [10]. For this reason, the theme of World Chagas Disease Day 2022 was "finding and reporting every case to defeat Chagas disease" [13].

Here we study the possibility of using the electrocardiogram (ECG) to screen for ChD. The ECG is a widely available, low-cost exam, often provided in primary care settings in endemic countries [14]. The automated analysis of ECG is a successful technology and has already improved the analysis of this exam over the past decades [15].

The field of artificial intelligence, in particular deep learning [16], has demonstrated promising performance for automated analysis. Besides the success of classifying common ECG diagnoses with high-performance [17, 18], the technology has presented successes in predicting and screening for diseases and diagnoses which traditionally were not directly possible only from the ECG. These include detection of myocardial infarction without ST-elevation [19], predicting the future development of atrial fibrillation from sinus rhythm exams [20, 21] and the ability to screen for cardiac contractile dysfunction [22]. Indeed, there is evidence that deep learning reading of ECGs detects more than traditional features, as is indicated by studies showing good prediction of age and even the risk of death [23–25].

In this study, we investigate whether a deep neural network can detect ChD and CCC from ECG tracings. Being able to evaluate ChD from this exam can help to detect cases in an early stage and enables early and more effective treatment.

## Methods

### Data sets

We develop our model using the SaMi-Trop data set [26] and the CODE data set [27]. The SaMi-Trop data set is a collection of ChD patients from the northern part of Minas Gerais, Brazil. The CODE data set [27] is more general, collected by the Telehealth Network of Minas Gerais (TNMG), Brazil [28]. For testing or external validation, we use the REDS-II data set [29] and the ELSA-Brasil data set [30]. The baseline characteristics of all four data sets are summarised in Tables 1 and 2.

**Definitions.**   Chronic ChD is diagnosed by the presence of two positive different serological tests against *T. cruzi* in both SaMi-Trop and REDS-II cohorts, as recommended by international guidelines [3]. In the ELSA-Brasil study, a cohort primarily designed to study chronic non-communicable diseases, the presence of Chagas disease was detected by the presence of only one positive serological test. In the CODE study, Chagas disease was self-reported by the patients since this electronic cohort is formed by patients under care in primary care units in the state of Minas Gerais. For SaMi-Trop, REDS-II and ELSA cohorts, ECGs were transmitted to an ECG reading center at the 'Centro de Telessaúde in Hospital

**Table 1. Development data sets baseline characteristics.** For CODE the Chagas patient reports their own condition, while for Samitrop the blood sample is used to determine the serological status. CCC stands for chronic Chagas cardiomyopathy, which is not available (n.a.) for the CODE data set. MI stands for myocardial infarction.

| | CODE (Patients, n = 1,556,767) | | SaMi-Trop (Patients, n = 2,054) | | |
| --- | --- | --- | --- | --- | --- |
| | **Not reported** | **Chagas** | **Seronegative** | **Chagas** | **CCC** |
| Patients, n | 1,524,766 | 32,001 | 144 | 1,910 | 1,111 |
| Sex (male), n (%) | 922,780 (39.7%) | 11705 (36.6%) | 60 (41.7%) | 629 (32.9%) | 398 (35.8%) |
| Age (years), mean (sd) | 52 (18) | 59 (14) | 64.0 (14.7) | 59.2 (12.9) | 60.4 (13.0) |
| Hypertension, n (%) | 482,184 (31.6%) | 21,849 (68.3%) | 112 (77.8%) | 1219 (63.8%) | 762 (68.6%) |
| Diabetes, n (%) | 102803 (6.7%) | 4,398 (13.7%) | 31 (21.5%) | 195 (10.2%) | 107 (9.6%) |
| Smoking, n (%) | 108,815 (7.0%) | 4,190 (13.1%) | 49 (38.9%)* | 49 (38.9%)* | 301 (29.2%)* |
| Previous MI, n (%) | 13,693 (0.9%) | 1,308 (4.1%) | 12 (8.3%) | 90 (4.7%) | 69 (6.2%) |
| Dislipidemia, n (%) | 62,686 (4.1%) | 3,823 (11.9%) | n.a. | n.a. | n.a |
| Obesity, n (%) | 86,500 (5.7%) | 3,321 (10.4%) | 24 (21.8%)* | 222 (15.0%)* | 106 (12.8%)* |
| Exams, n | 2,257,122 | 47,474 | 365 | 4,654 | 2,693 |

* For CODE, comorbidity data is self-reported and might be underrepresented. There is missing data regarding smoking (seronegative=18/chagas=148/ccc=81 missing entries) and obesity (34/431/281 missing entries) in the SaMi-Trop data set. For these entries, we report the percentages that consider the total of patients without missing data. Smoking gives the current smoking habits of the patient. Dislipidemia is not available (n.a.) for SaMi-Trop data set.

das Clínicas' in Belo Horizonte, Minas Gerais for standardized measurement, reporting and codification according to the Minnesota coding criteria in a validated ECG data management software [31]. Major ECG abnormalities were considered according to standard definitions [32], and all tracings with a major ECG abnormality have been reviewed by an experienced cardiologist.

**CODE.** The Clinical Outcomes in Digital Electrocardiography (CODE) data set was developed with the database of digital ECG exams of the TNMG and a detailed description of the cohort can be obtained at [27]. The data set was collected between 2010 and 2017 from 811 counties in the state of Minas Gerais, Brazil. A subset of 15% of this data set is available online [33].

**Table 2. Test data sets baseline characteristics.** Blood sample is used to determine the serological status in both datasets.

| | REDS-II (Patientes, n = 631) | | | ELSA-Brasil (Patientes, n = 13,739) | | |
| --- | --- | --- | --- | --- | --- | --- |
| | **Seronegative** | **Chagas** | **CCC** | **Seronegative** | **Chagas** | **CCC** |
| Patients, n | 283 | 348 | 149 | 13,459 | 280 | 46 |
| Sex (male), n (%) | 140 (49.5%) | 171 (49.1%) | 82 (55.0%) | 167 (59.6%) | 19 (41.3%) | 6,256 (45.5%) |
| Age (years), mean (sd) | 58.2 (9.6) | 56.1 (9.8) | 56.4 (9.8) | 52.1 (9.2) | 57.4 (9.1) | 61.1 (6.8) |
| Hypertension, n (%) | 101 (35.7%)* | 136 (39.1%)* | 62 (41.6%)* | 4,814 (35.8%)* | 135.0 (48.2%)* | 27.0 (58.7%)* |
| Diabetes, n (%) | 38 (13.4%)* | 45 (12.9%)* | 16 (10.7%)* | 2,160 (16.0%) | 57.0 (20.4%) | 15 (32.6%) |
| Smoking, n (%) | 94 (33.2%) | 98 (28.2%) | 42 (28.2%) | 1 769 (13.1%) | 32 (11.4%) | 4 (8.7%) |
| Previous MI, n (%) | 9 (3.2%)* | 13 (3.7%)* | 8 (5.4%)* | 242 (1.8%) | 9 (3.2%) | 5.0 (10.9%) |
| Dislipidemia, n (%) | n.a. | n.a. | n.a. | 8,853 (65.8%) | 201 (71.8%) | 36 (78.3%) |
| Obesity, n (%) | 68 (24.6%)* | 66 (19.4%)* | 28 (19.0%)* | 3,079 (22.9%) | 67 (23.9%) | 10 (21.7%) |
| Exams, n | 283 | 348 | 149 | 13,459 | 280 | 46 |

*There is missing data regarding regarding Hypertension (12/9/3 missing entries), Diabetes (2/8/4 missing entries), Previous MI (0/5/1 missing entries) and Obesity (7/8/2 missing entries) in the REDS-II data set; and, regarding Hypertension (16/0/0 missing entries) in the ELSA-Brasil. For these entries, we report the percentages that consider the total of patients without missing data. Dislipidemia is not available (n.a.) for REDS-II data set.

From an initial data set of 2,470,424 ECGs, 1,773,689 patients were identified. This initial data set contains the SaMi-Trop data set. Therefore, we first remove the patients from the SaMi-Trop study to avoid any overlap. Additionally, we have to exclude the ECGs with technical problems and those from patients under age 16, resulting in a total of 2,304,596 ECG records from 1,556,767 patients.

In this data set, the labels of ChD rely on self-reported diagnoses during the consultation. A total of 47, 474 ECGs (2.0%) from 25, 252 patients (1.6%) are labelled as positive ChD cases. The serological status of the self-reported Chagas labels has not been checked, and it is also unclear whether the patient has already developed CCC or not.

**SaMi-Trop.**   The study was conducted through a collaboration between scientists within the São Paulo-Minas Gerais Tropical Medicine Research Center (SaMi-Trop), formed with a specific research focus on ChD. [34] The study selected eligible patients with self-reported ChD diagnosis. This data set was collected in 21 Brazilian municipalities from ECGs taken between 2010 and 2012 by the TNMG. The connection to the TNMG explains the intersection of the SaMi-Trop data set with the CODE data set. The study has a follow-up time of two years. It is partially available in [35]

A total of 2, 157 patients were assessed in the study. Among the patients from the original SaMi-Trop study, we removed 22 patients with an undefined serological status, and the remaining 83 for not having a paired ECG recording. After the exclusions, the resulting data set comprises 2, 054 patients with 1, 910 ChD positive patients (93.4%). The positive patients consist of 1, 111 patients with CCC (54.1% of total sample) and 799 without (38.9% of total sample).

Some of the patients have taken multiple ECG recordings during an exam which we utilize during development as a form of data augmentation. Hence, we have 5, 019 SaMi-Trop ECG traces available including 2, 693 traces with CCC (53.7%) and 1, 961 traces without (39.1%).

**REDS-II.**   The Retrovirus Epidemiology Donor Study-II (REDS-II) data set was collected to observe the natural history of ChD patients in São Paulo and Montes Carlos, Brazil from blood donors. Seropositive and seronegative patients examined in 1996–2002 were re-examined in 2008–10 [4] with ECG exams and again in 2018–19 [29]. The data set consists of 631 patients that performed an ECG in the last visit in 2018–19, including 348 ChD patients (55.8%), of which 149 patients had CCC (23.6% of the total sample). The model is evaluated using a single exam from each patient (the first one).

**ELSA-Brasil.**   The Brazilian Longitudinal Study of Adult Health (ELSA-Brasil) aimed to examine risk factors and the long-term incidence of chronic diseases with focus on cardiovascular diseases and diabetes. The baseline evaluation was performed in 2008–2010 and recruited active and retired civil servants from five universities and research institutes from 6 different Brazilian states. ChD serological status and standardized ECG were obtained from all participants [36, 37].

The data set consists of 15,105 patients in total. We remove 27 patients where the ChD serological status is not available, 12 patients where the serological status is inconclusive, and 1,327 patients from which the ECG traces are not available. After the exclusions, we have a data set with a total of 13,739 patients. ChD was confirmed in 280 of the patients (2.0%), of which 46 had CCC (0.3% of the total sample). The model is evaluated using a single exam from each patient (the first one).

## Model

**Data preprocessing.**   The ECG signals have been re-sampled such that all ECGs have the same sampling frequency of 400 Hz. Each input ECG has 4, 096 time samples for each of the 12 standard ECG leads. Original signals of a shorter time span have been extended through

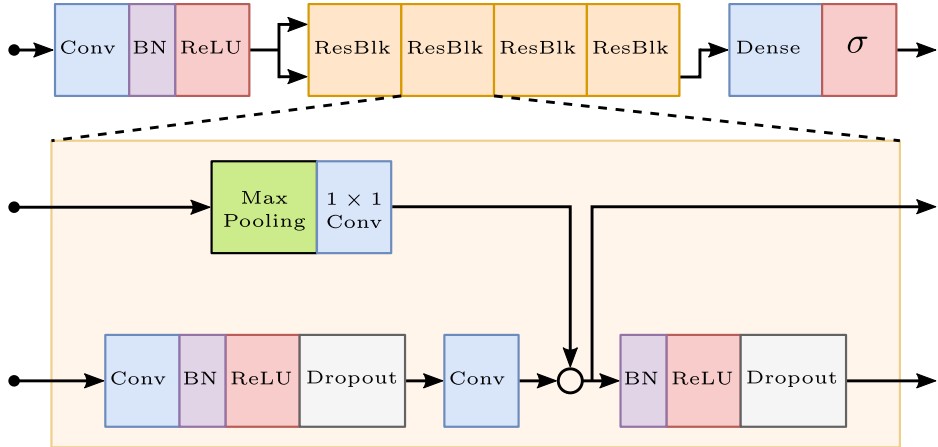

**Fig 1. Network architecture.** The figure was originally illustrated in [17].

zero-padding. The output data comprises binary scalar variables corresponding to positive or negative diagnose. We combine positive cases with and without CCC in our model in order to focus on the class of positive ChD cases in general.

**Architecture.** The deep learning model consists of a residual neural network (ResNet) adapted to uni-dimensional signals, and includes convolutional layers both before and within the residual blocks. Our network architecture is visualised in Fig 1. We make use of the same network architecture as [17], where the CODE data set was utilised to classify multiple ECG abnormalities; we refer to that work for further details and note that we have modified the final output layer in adaptation to our binary classification. The model is implemented in PyTorch [38], building upon code used in related work [39, 40].

**Parameter tuning.** The learnable parameters of the neural network are chosen through minimisation of the binary cross-entropy loss function. For increased computational efficiency, we split the training data into mini-batches of size 32.

We use both the CODE and SaMi-Trop data sets during the training phase. This way, we utilise the size of the CODE data set—with many examples of negative diagnoses—as well as the high-quality (mainly positive) entries of SaMi-Trop. Both data sets contribute with 50% of the data that the model experience in each mini-batch. The validation data is an independent mix of 30% of the SaMi-Trop entries and twice as many entries from CODE.

The dropout rate is 0.5, and we use a weight decay of 0.001 to reduce the risk of overfitting. The learning rate is initially set to 0.001 and is decreased in a step-wise manner by a factor 10 when the validation loss has not decreased for ten subsequent epochs (counted with respect to SaMi-Trop)—we terminate the optimisation if the learning rate drops below $10^{-7}$. We apply early stopping by using the network parameter values associated with the lowest validation loss for testing.

To reduce the sensitiveness of the weight initialisation, we use an ensemble approach by running the optimisation 15 times with different random seeds, and then averaging the outputs of the final models. The progression of the losses evaluated on the training and validation data sets are displayed in Fig 2.

**Threshold selection.** The model output is a value between 0 and 1 and can loosely be interpreted as the predicted probability of ChD being present in the exam analysed. The Chagas diagnose is predicted as positive when the model output is above a given classification threshold.

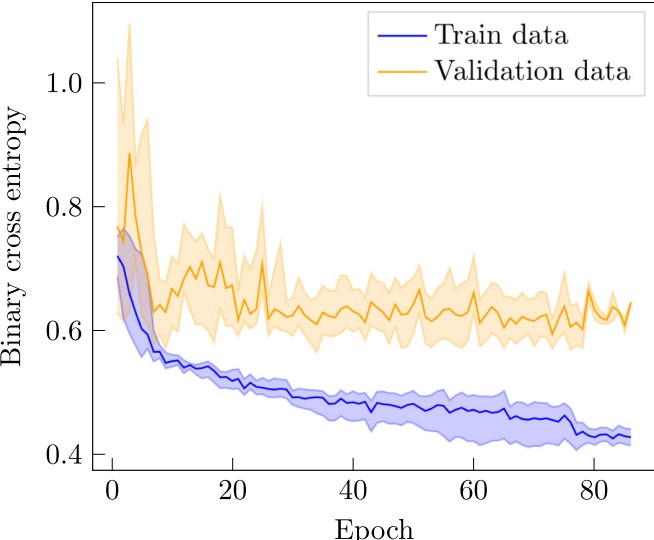

**Fig 2. Loss function evaluation.** The shaded regions correspond to the maximum and minimum values of 15 separate learning processes with different weight initialisations. The solid lines are the averages.

We consider two different approaches to selecting the threshold. The first one is by maximising the F1 score (i.e. the harmonic mean of precision and recall) on the validation data. This threshold is suitable for balanced or moderately imbalanced data sets where the main interest is to diagnose the patients under consideration.

The second approach is to choose the threshold by requiring a certain specificity on the validation data. The higher the specificity, the more likely is the model to correctly diagnose a negative patient. As a high specificity typically is desired for screening purposes, this approach for threshold selection is motivated on highly imbalanced data sets (which reflects the Chagas prevalence in the population as a whole).

The first approach is used on the REDS-II test set since this data set is only moderately imbalanced (55.8% ChD and 23.6% CCC ECGs). On the ELSA-Brasil test set the threshold is selected according to the second approach since this data set is more imbalanced (2.0% ChD and 0.3% CCC ECGs). We select the threshold by requiring a 90% specificity on the validation data.

## Evaluation

**Metrics.** Recall (also known as sensitivity), specificity and precision are threshold-dependent metrics that we used to evaluate and report the model performance. Recall or sensitivity specifies the ratio of true positive predictions to positive cases (i.e. the ratio of the positive cases that are indeed predicted as positive); specificity denotes the ratio of true negative predictions to negative cases; and precision is the ratio of true positive predictions to all positive predictions (the ratio of all positive predictions that are correct).

We also report two threshold-independent metrics. The AUC-ROC (also known as c-statistics) is the integral of the receiver-operator characteristics (ROC), and can be interpreted as the probability that a randomly chosen sample with positive label is assigned a higher output than a randomly chosen sample with negative label. Lastly, we report the average precision, which is obtained by integrating the precision-recall curve and thereby summarising it into a single value.

**Analysis of the results in groups.** As part of the model analysis we evaluate the model performance in different subgroups of patients. We stratify the patients by age group {16–40, 40–49, 50–59, 60–69, 70+} and sex {male, female}. Bootstrapping [41] is used to analyse the empirical distribution of the metrics in each subgroup. We generate 1, 000 different data sets by sampling with replacement from the test set (each with the same number of samples as in the test set). Using the bootstrapped data sets, we compute the evaluation metrics described above and present the results in box plots.

**Visualisation tools.** To identify possible patterns in the classification, we highlight parts of the ECG that the model focuses on for its prediction using an adaptation of the Grad-CAM visualisation method [42]. Visualisations are generated in two steps: in a forward pass we compute the activations of the neural network in an intermediary layer (we use the first convolutional layer of the first residual block), and in a backward pass we compute the gradients corresponding to these activations. The gradients are averaged to get the relative importance of each channel, which is then used to compute a proportional mean of the activations.

In essence, these plots highlight which parts of the ECG the network assigns particularly high importance. We generated the Grad-CAM plots for 20 cases (10 with CCC and 10 without) with the highest probability among the true positive cases. These plots were then inspected and analysed by a cardiologist for possible medical patterns.

## Results

We evaluated the model performance on the validation data and the external test data sets. The ROC curve performance is displayed in Fig 3. The model attains AUC-ROC values of 0.80 (CI 95% 0.79–0.82) for the validation data set, 0.68 (CI 95% 0.63–0.71) for REDS-II and 0.59 (CI 95% 0.56–0.63) for ELSA-Brasil. The confidence intervals have been formed by bootstrapping the output of the ensemble model. Table 3 lists all performance metrics evaluated on the validation data for two different thresholds selected through the aforementioned approaches. The same metrics evaluated on the test data sets are listed in Table 4. Additionally, we also analysed the precision-recall curve and the empirical probabilities predicted by the model. These

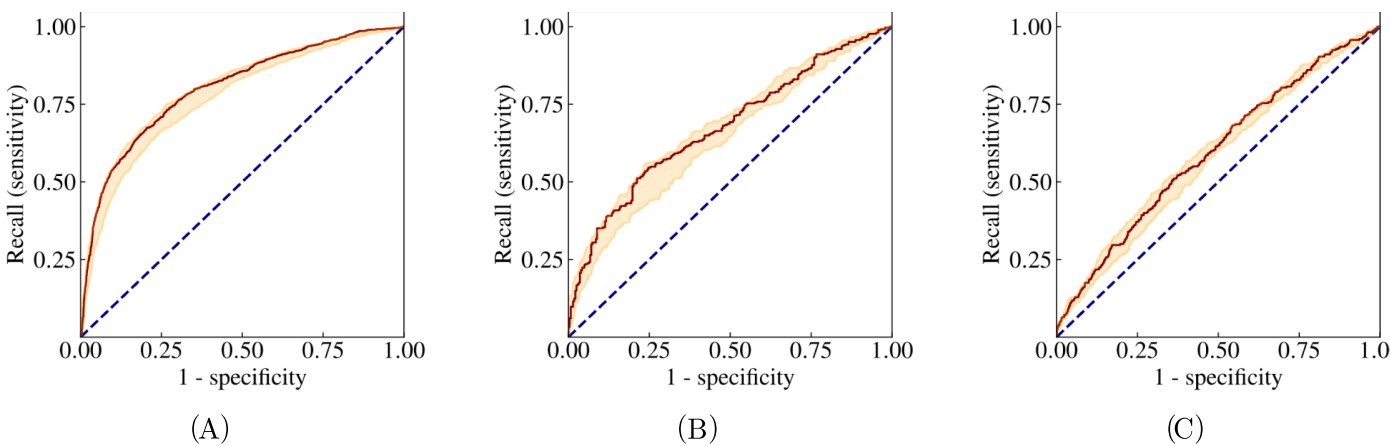

**Fig 3. ROC curves on validation and test data: ChD vs normal.** Receiver operating characteristics (ROC) computed on the validation data set (SaMi-Trop + CODE) (a) and the test data sets, REDS-II (b) and ELSA-Brasil (c). The shaded regions encapsulate the maximum and minimum values corresponding to 15 different weight initialisations. The outputs of the 15 trained models are averaged to produce the output of the ensemble model, the result of which is shown by the solid lines. The 95% CI is obtained by bootstrapping the ensemble model. The dotted blue lines correspond to completely random assignment of class probabilities.

**Table 3. Results on validation data.** Metrics and 95% confidence intervals evaluated on the validation data set for two different classification thresholds: 0.60 (selected by maximising the F1 score) and 0.71 (corresponding to 90% specificity).

| Metric \ Threshold | F1 score-based: 0.60 | Specificity-based: 0.71 |
|---|---|---|
| Recall | 0.67 (0.65–0.70) | 0.54 (0.51–0.57) |
| Specificity | 0.80 (0.78–0.81) | 0.90 (0.89–0.91) |
| Precision | 0.62 (0.59–0.64) | 0.73 (0.70–0.75) |
| F1 score | 0.64 (0.62–0.66) | 0.62 (0.60–0.64) |
| AUC-ROC | 0.80 (0.79–0.82) | |
| Average precision | 0.68 (0.65–0.70) | |

results are displayed in S1 and S3 Figs. The metrics for subgroups stratified by age and sex are displayed in Fig 4.

We also evaluated the model for considering only patients with CCC as positive. In this case, the model attains an AUC-ROC of 0.82 (CI 95% 0.77–0.86) for REDS-II and 0.77 (CI 95% 0.68–0.85) for ELSA-Brasil (see Fig 5). All metrics for this configuration are included in Table 4. We also analysed the precision-recall curve and the empirical probabilities predicted by the model (S2 and S4 Figs).

In S5 Fig and S1 Table, we show the additional results for another test set configuration. Namely where the patients with CCC have been excluded; the remaining patients where ChD was detected are here constituting the positive cases (this configuration is indicated "no CCC"). We also show the result of a model trained to detect CCC (with all others being considered negative): S6 Fig shows the training curve, S7 Fig shows the ROC curves, precision-recall curves and empirical distribution of the probabilities, and finally, S1 and S2 Tables give the performance metrics in this case.

The Grad-CAM analysis is presented in Fig 6, which shows three representative leads of a patient with CCC from the ELSA-Brasil data set. The shaded regions illustrate what parts of the signals the model considers to be of particular importance for the prediction. In S8 Fig we include the equivalent plots for another three patients with positive Chagas diagnose, one with and two without CCC.

## Discussion

Deep neural network-enabled analysis of the ECG is a topic of intense research [19–25]. Such methods have shown promising potential in detecting diverse conditions that are not traditionally diagnosed from the ECG, such as contractile disfunction [22] or non-STEMI myocardial infarction [19]. ChD is the parasitic disease with the most impact in South America [43] and it affects the lives of millions of individuals worldwide. Early detection of this disease can

**Table 4. Results on test data.** Metrics and 95% confidence intervals were evaluated on two different configurations of the test data sets. Left we consider ChD and CCC as positive. Right we only consider CCC as positive. The classification thresholds are 0.60 for REDS-II and 0.71 for ELSA-Brasil.

| Metric \ Test data | REDS-II | ELSA-Brasil | REDS-II (CCC) | ELSA-Brasil (CCC) |
|---|---|---|---|---|
| Recall | 0.52 (0.47–0.57) | 0.36 (0.30–0.42) | 0.79 (0.72–0.85) | 0.70 (0.56–0.82) |
| Specificity | 0.77 (0.72–0.81) | 0.76 (0.75–0.77) | 0.73 (0.69–0.76) | 0.76 (0.75–0.77) |
| Precision | 0.73 (0.68–0.79) | 0.03 (0.02–0.04) | 0.47 (0.41–0.53) | 0.01 (0.01–0.01) |
| F1 score | 0.61 (0.57–0.66) | 0.06 (0.05–0.07) | 0.59 (0.53–0.64) | 0.02 (0.01–0.03) |
| AUC-ROC | 0.68 (0.63–0.71) | 0.59 (0.56–0.63) | 0.82 (0.77–0.86) | 0.77 (0.68–0.85) |
| Average precision | 0.74 (0.69–0.78) | 0.04 (0.03–0.06) | 0.69 (0.61–0.76) | 0.10 (0.03–0.19) |

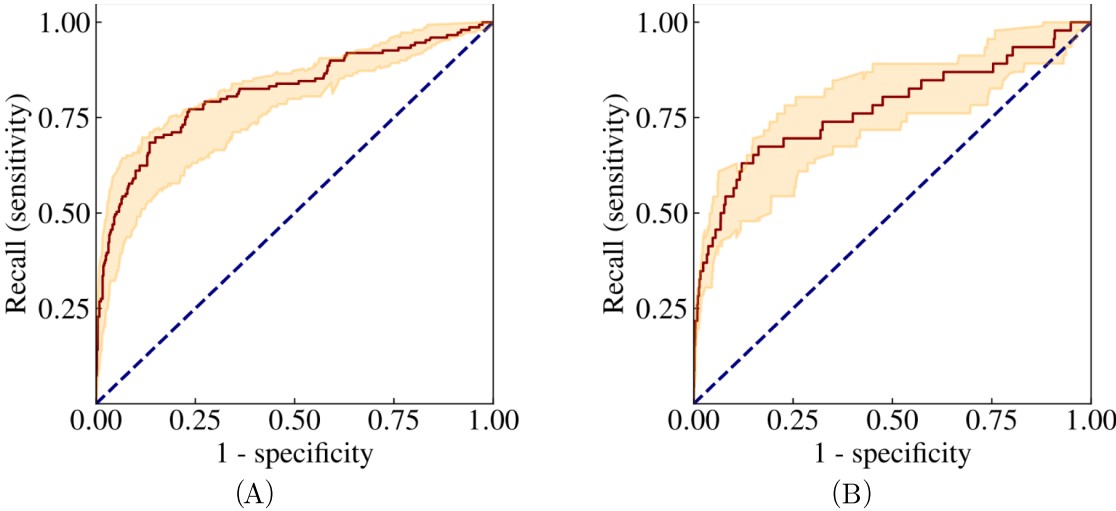

**Fig 4. Results stratified by subgroup.** Box plots of the model performance on the REDS-II (top row) and ELSA-Brasil (bottom row) test sets stratified by age (left column) and sex (right column). The box plots give the performance on 1000 bootstrapped samples.

therefore have a huge impact. Antiparasitic drugs are most effective in the early stage of the disease, however, most patients only become aware that they are infected much later when the patient is already in the later stage of the disease and presents other manifestations. Providing early treatment and the usage of advanced artificial intelligence or machine learning methods

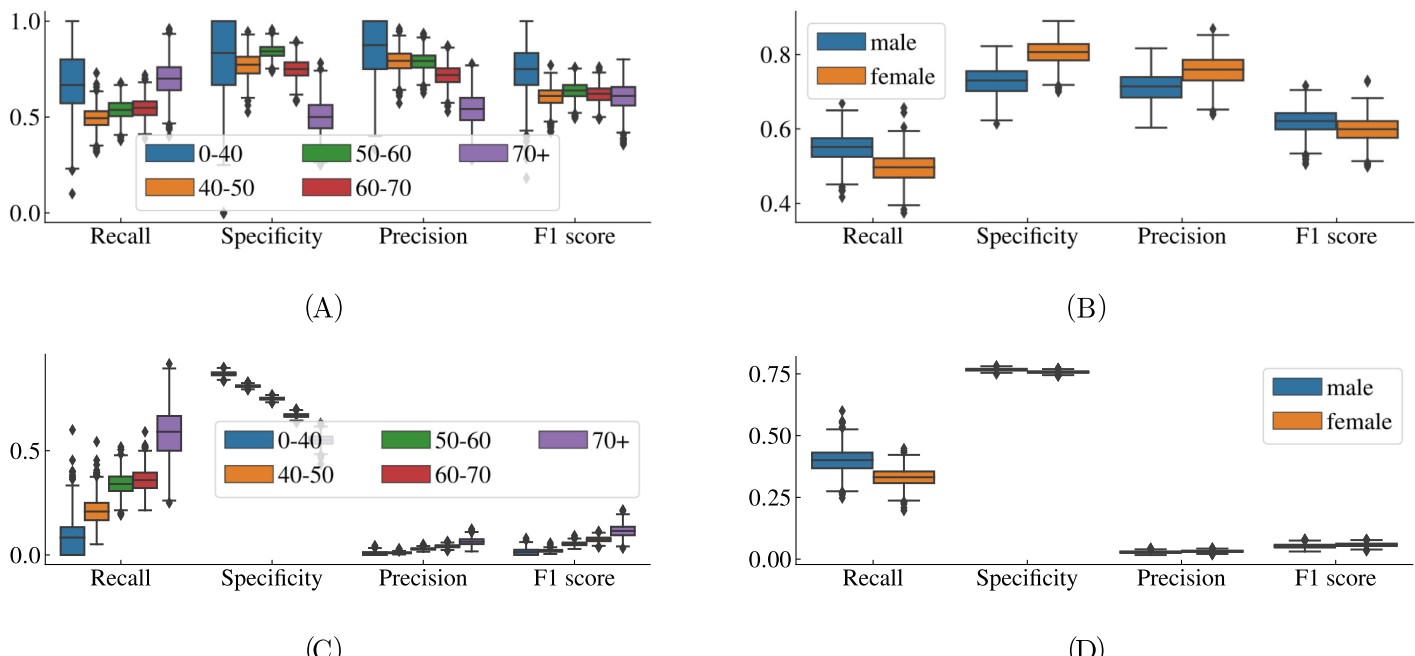

**Fig 5. ROC curves on test data: CCC vs all.** Receiver operating characteristics computed in REDS-II and ELSA-Brasil for predicting Chagas Cardiomiopathy. The shaded regions encapsulate the maximum and minimum values corresponding to 15 different weight initialisations—the outputs of these models are averaged to produce the output of the ensemble model, the result of which is given by the solid lines. The dotted blue lines correspond to completely random assignment of class probabilities.

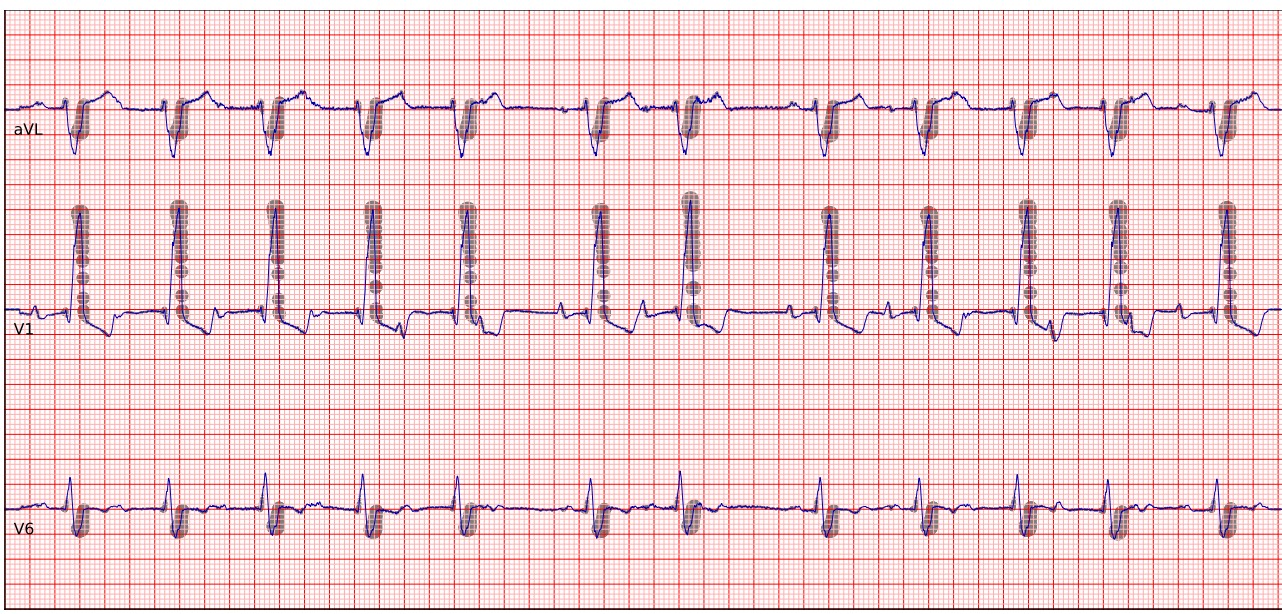

**Fig 6. Grad-CAM analysis.** Grad-CAM plot for a patient with CCC from the ELSA-Brasil data set, correctly classified by the model as Chagas positive. This plot includes three representative leads (top to bottom: aVL, V1 and V6). The shading indicates regions that the model assigns particular importance for its prediction.

for the detection of this disease presents itself as a promising alternative. To the best of the authors' knowledge, this is the first study to present such an application.

The development of data-driven methods for automatic diagnosis of neglected diseases presents a challenge of its own. These diseases usually affect areas where the population is underprivileged and have little access to the healthcare system. The data might not come in well-organised databases or might not even be stored in electronic format. In this sense, the CODE, SaMi-Trop, ELSA-Brasil and REDS-II cohorts are extremely valuable: they are medium or large-size and well-kept data sets that can be used for developing and testing such tools.

The results we present are promising and indicate that the model is capable of detecting patients with CCC from the ECG tracings with high discrimination. For patients without CCC, the discrimination is lower.

In light of the results, it is natural to ask if we can further improve the performance with respect to patients with CCC. Therefore, we restrict the positive diagnoses to patients with CCC during the training phase and consider all patients without CCC as negatives (this implies that ChD positive patients without CCC are considered negative in this scenario). The result of this approach is given in S1 and S2 Tables. All metrics considered, except for the recall, are indeed improved. Thus, this model might be the preferable choice for CCC detection.

Chagas cardiomyopathy is characterised by a group of typical ECG abnormalities, frequently combining conduction disturbances, especially right bundle branch block with left anterior hemiblock, associated with rhythm disorders, such as ventricular ectopic beats and atrial fibrillation [44–46]. Thus, it is unsurprising that our Grad-CAM analysis depicts exactly the late portion of the QRS in cases with a bundle branch block. It is interesting that the Grad-CAM map also depicts the QRS complex when recognising the ChD patients with CCC, maybe related to the presence of high frequency, low amplitude abnormalities typical of

fibrosis, which can occur early in the natural history of ChD [47]. However, this type of analysis has clear limitations [48, 49] since heatmaps can provide information on where the critical area for the neural network model is to make a decision but not inform if the abnormality is related to changes in voltage, duration or morphology modification of the ECG tracing. Moreover, recurrent features, like the RR interval, are not shown in this kind of heatmaps. Our analysis here is also limited to a small set of correct model predictions and does not represent a statistical analysis. Hence, we cannot deduct general rules for the diagnosis of ChD but we can identify from the unsurprising areas where the model focuses on that it does not use some unrelated proxy information to make its predictions.

Comparing the two test data sets, we obtain similar performance for discrimination in terms of AUC-ROC, but very different precision. This indicates that our model predicts many false positives for the ELSA-Brasil data set. Given the vast difference in prevalence for ChD patients in ELSA-Brasil (2.0%) and REDS-II (55.1%), it is reasonable that for ELSA-Brasil our model will by default have lower precision. We can also observe the large portion of false positive cases in S3 Fig panel C when choosing a threshold of 0.60 (based on F1 score) or even 0.71 (based on 90% specificity). We believe the performance could be improved with the addition of epidemiological questions, and that our model can be a useful tool in helping pre-selecting patients for further testing in order to determine the infection with ChD.

As previously mentioned, the ChD status in the CODE data set is based on self-reporting by the patients, and the labels are thus suffering from notable uncertainty. Thus, testing on these labels might be uninformative and we have used more reliable databases such as ELSA-Brasil and REDS-II to get a better estimate of our model performance. Nonetheless, the labels in CODE still contain a sufficient amount of information to learn about CCC patients and the data set was indeed useful in developing a better-performing model. Methods designed to reduce the impact of label noise (see e.g. [50, 51]) could potentially be employed for more efficient use of the CODE data.

Our model could be even more insightful if we could test it on other openly available data sets. However, data sets about neglected diseases are scarce and both ELSA-Brasil as well as REDS-II are valuable but also medium to large-scale sources to rigorously test the model. Furthermore, a comparison with other models or software for Chagas detection would be useful, but unfortunately, it is not possible—to the best of our knowledge, this is the first work that tackles automatic diagnosis of Chagas directly from the ECG. Therefore, this study serves as a first baseline that opens a new line of work for further improvements.

Our findings are particularly valuable under the scantiness of validated strategies to detect ChD patients in endemic regions. Current recommendations for screening include all patients who were born in or have lived for an extended period in ChD endemic zones [44], which can be challenging, especially in endemic countries, since it can encompass the whole population of a region. A risk score was developed specifically to answer the question, "Does my patient have chronic Chagas disease?" but it seems to have limited practical value since it includes 13 variables obtained from clinical and epidemiological history and from a conventionally analysed 12-lead ECG [52]. It implies that the best approach would merge conventional and non-conventional methods [53], including the use of rapid point-of-care serological tests [54].

A clinical study would be particularly valuable, as the performance of the model could be evaluated directly by clinicians and patients. At this stage, we foresee our model as a pre-selection method of patients for further screening of the serological status. It is important to underline that more available data will enable improvements of the model that can be adapted into its daily clinical practice. We hope that a future study will evaluate the clinical relevance of our model to improve the early diagnosis of ChD.

## Supporting information

**S1 Table. Results on test data: No-CCC configuration.** Metrics and 95% confidence intervals evaluated on the no-CCC configuration of the test data sets (see the text for details). The classification thresholds are 0.60 for REDS-II and 0.71 for ELSA-Brasil.
(XLSX)

**S2 Table. Results on validation data: CCC-specific training.** Equivalent to Table 3 when the training is adapted to specifically target patients with CCC. The classification thresholds are 0.51 (selected by maximising the F1 score) and 0.33 (corresponding to 90% specificity).
(XLSX)

**S3 Table. Results on test data: CCC-specific training.** Equivalent to Table 4 when the training is adapted to specifically target patients with chronic Chagas cardiomyopathy. The classification thresholds are 0.51 for REDS-II and 0.33 for ELSA-Brasil.
(XLSX)

**S1 Fig. Precision-recall curves on validation and test data: ChD+CCC vs normal.** The shaded regions encapsulate the maximum and minimum values corresponding to 15 different weight initialisations—the outputs of these models are averaged to produce the output of the ensemble model, the result of which is given by the solid lines.
(EPS)

**S2 Fig. Precision-recall curves on test data: CCC vs rest.** We consider only CCC as positive and ChD as well as normal as negative here.
(EPS)

**S3 Fig. Output histograms: ChD+CCC vs normal.** Histograms are computed on the validation data and the test data. Note the logarithmic scale of the $y$-axis. We can see the number of false positive/negatives when applying the selected thresholds on the $x$-axis: 0.60 for REDS-II and 0.71 for ELSA-Brasil.
(EPS)

**S4 Fig. Output histograms: CCC vs rest.** Histograms are computed on the test data where we only consider CCC as positive and ChD as well as normal as negative. Note the logarithmic scale of the $y$-axis.
(EPS)

**S5 Fig. Results on test data: No-CCC configuration.** Receiver operating characteristics (left), precision-recall curves (middle) and output histograms (right) computed on the test data for the no-CCC configuration (see the text for details). This set removed the CCC cases and shows ChD vs normal.
(EPS)

**S6 Fig. Loss function evaluation: CCC-specific training.** Equivalent to Fig 2 when the training is adapted to specifically target patients with CCC.
(EPS)

**S7 Fig. Results on test data: CCC-specific training.** Receiver operating characteristics (left), precision-recall curves (middle) and output histograms (right) computed on the validation data and the test data when the training is adapted to specifically target patients with CCC.
(EPS)

**S8 Fig. Grad-CAM analysis: Additional patients.** Complementing Fig 6 with another three Grad-CAM plots for patients from the ELSA-Brasil data set, correctly classified by the model

as Chagas positive. We here include one patient with chronic Chagas cardiomyopathy (a), and two without (b-c). The plots include three representative leads (top to bottom: aVL, V1 and V6). The shading indicates regions that the model assigns particular importance.
(EPS)

## Author Contributions

**Conceptualization:** Carl Jidling, Antonio L. P. Ribeiro, Antônio H. Ribeiro.

**Data curation:** Claudia Di Lorenzo Oliveira, Clareci Silva Cardoso, Ariela Mota Ferreira, Luana Giatti, Sandhi Maria Barreto, Ester C. Sabino, Antonio L. P. Ribeiro, Antônio H. Ribeiro.

**Formal analysis:** Carl Jidling, Daniel Gedon, Antônio H. Ribeiro.

**Funding acquisition:** Thomas B. Schön, Sandhi Maria Barreto, Ester C. Sabino, Antonio L. P. Ribeiro.

**Investigation:** Carl Jidling, Daniel Gedon, Thomas B. Schön, Antonio L. P. Ribeiro, Antônio H. Ribeiro.

**Methodology:** Carl Jidling, Daniel Gedon, Thomas B. Schön, Antonio L. P. Ribeiro, Antônio H. Ribeiro.

**Project administration:** Thomas B. Schön, Antonio L. P. Ribeiro, Antônio H. Ribeiro.

**Resources:** Carl Jidling, Daniel Gedon, Antônio H. Ribeiro.

**Software:** Carl Jidling, Daniel Gedon, Antônio H. Ribeiro.

**Supervision:** Thomas B. Schön, Antonio L. P. Ribeiro, Antônio H. Ribeiro.

**Validation:** Carl Jidling, Daniel Gedon.

**Visualization:** Carl Jidling, Daniel Gedon.

**Writing – original draft:** Carl Jidling, Daniel Gedon, Thomas B. Schön, Antonio L. P. Ribeiro, Antônio H. Ribeiro.

**Writing – review & editing:** Carl Jidling, Daniel Gedon, Thomas B. Schön, Claudia Di Lorenzo Oliveira, Clareci Silva Cardoso, Ariela Mota Ferreira, Luana Giatti, Sandhi Maria Barreto, Ester C. Sabino, Antonio L. P. Ribeiro, Antônio H. Ribeiro.

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
