## [Decision Letter · Decision Letter 0]

25 May 2023

Dear Dr. Ribeiro,

We are pleased to inform you that your manuscript 'Screening for Chagas disease from the electrocardiogram using a deep neural network' has been provisionally accepted for publication in PLOS Neglected Tropical Diseases.

Best regards,

Luisa Magalhães

Academic Editor

Ricardo Fujiwara

Section Editor

Reviewer's Responses to Questions

**Key Review Criteria Required for Acceptance?**

**Methods**

-Are the objectives of the study clearly articulated with a clear testable hypothesis stated?

-Is the study design appropriate to address the stated objectives?

-Is the population clearly described and appropriate for the hypothesis being tested?

-Is the sample size sufficient to ensure adequate power to address the hypothesis being tested?

-Were correct statistical analysis used to support conclusions?

-Are there concerns about ethical or regulatory requirements being met?

Reviewer #1: -Are the objectives of the study clearly articulated with a clear testable hypothesis stated? Yes

-Is the study design appropriate to address the stated objectives? Yes

-Is the population clearly described and appropriate for the hypothesis being tested?Yes

-Is the sample size sufficient to ensure adequate power to address the hypothesis being tested? Partially Yes

-Were correct statistical analysis used to support conclusions? Yes

-Are there concerns about ethical or regulatory requirements being met? Not

Reviewer #2: Must review all the document with a native english speaker, lots of minimum language details.

**Results**

-Does the analysis presented match the analysis plan?

-Are the results clearly and completely presented?

-Are the figures (Tables, Images) of sufficient quality for clarity?

Reviewer #1: Results

-Does the analysis presented match the analysis plan? Yes

-Are the results clearly and completely presented? Yes

-Are the figures (Tables, Images) of sufficient quality for clarity? Yes

Reviewer #2: (No Response)

**Conclusions**

-Are the conclusions supported by the data presented?

-Are the limitations of analysis clearly described?

-Do the authors discuss how these data can be helpful to advance our understanding of the topic under study?

-Is public health relevance addressed?

Reviewer #1: -Are the conclusions supported by the data presented? Yes

-Are the limitations of analysis clearly described? Yes

-Do the authors discuss how these data can be helpful to advance our understanding of the topic under study? Yes

-Is public health relevance addressed? Yes

Reviewer #2: (No Response)

**Editorial and Data Presentation Modifications?**

Reviewer #1: Dear Editor,

I have carefully reviewed the manuscript entitled: “Screening for Chagas disease from the electrocardiogram using a deep neural network”, which presents a deep learning model based on a residual neural network architecture for detecting Chagas disease (ChD) and Chronic Chagas Cardiomyopathy (CCC) from electrocardiogram (ECG) signals. The authors have used multiple cohorts to train, validate and test their model and have presented the results of their evaluation. Although the performance was not particularly exceptional in the validation cohorts, this represents a major step forward in the effort to design tools that facilitate the objective assessment of Chagas disease risk in an individual from an endemic area. Overall, the article is optimally written and does not present major limitations that would preclude its publication in the journal. I have only a few minor comments to make.

- The decision-making process of Deep learning models is difficult to understand due to their “black box” nature. The authors nicely address this limitation and improve its understandability by using the Grad-CAM visualization method. However, ¿could we extract the most relevant features used by the model and their weights in order to be able of applying them without the need of the algorithm? This considering that a significant proportion of physicians assessing CD patients may not have access to the whole model or the possibility of applying it without an app.

-¿Are there potential barriers to clinical adoption of the proposed model? (need for high-quality digital ECG data, potential resistance from clinicians, and how the model would fit into the existing diagnostic workflow, among others). This could be considered a limitation as it stands between the research and its potential impact on real-world clinical practice.

-The authors observed better performance when evaluating only patients with cardiomyopathy as cases. Are ejection fraction data available for a subgroup of the population? It is of great interest to me whether this method can be used for heart failure patients with reduced ejection fraction secondary to CCC because, although it is hard to believe, serological testing for Chagas disease is not available in many Latin American heart failure centers. It is likely that the performance will be further improved to differentiate patients with heart failure secondary to CCC vs other etiologies.

Reviewer #2: (No Response)

**Summary and General Comments**

Reviewer #1: Overall, the article is optimally written and does not present major limitations that would preclude its publication in the journal.

Reviewer #2: (No Response)

PLOS authors have the option to publish the peer review history of their article (what does this mean?). If published, this will include your full peer review and any attached files.

Reviewer #1: No

Reviewer #2: **Yes: **Ezequiel José Zaidel

---

## [Editor Report · Acceptance letter]

27 Jun 2023

Dear Dr. Ribeiro,

We are delighted to inform you that your manuscript, "Screening for Chagas disease from the electrocardiogram using a deep neural network," has been formally accepted for publication in PLOS Neglected Tropical Diseases.

Best regards,

Shaden Kamhawi

co-Editor-in-Chief

Paul Brindley

co-Editor-in-Chief
